# Distinct population and single-neuron selectivity for executive and episodic processing in human dorsal posterior cingulate

Lyndsey Aponik-Gremillion[1,2,3], Yvonne Y Chen[4], Eleonora Bartoli[3], Seth R Koslov[4], Hernan G Rey[3,5,6], Kevin S Weiner[7], Daniel Yoshor[4], Benjamin Y Hayden[8,9,10], Sameer A Sheth[1,3], Brett L Foster[4]*

[1]Department of Neuroscience, Baylor College of Medicine, Houston, United States; [2]Department of Health Sciences, Dumke College for Health Professionals, Weber State University, Ogden, United States; [3]Department of Neurosurgery, Baylor College of Medicine, Houston, United States; [4]Department of Neurosurgery, Perelman School of Medicine, University of Pennsylvania, Philadelphia, United States; [5]Department of Neurosurgery, Medical College of Wisconsin, Milwaukee, United States; [6]Joint Department of Biomedical Engineering, Medical College of Wisconsin, Milwaukee, United States; [7]Department of Psychology, University of California, Berkeley, Berkeley, United States; [8]Department of Neuroscience, University of Minnesota, Minneapolis, United States; [9]Center for Magnetic Resonance Research, University of Minnesota, Minneapolis, United States; [10]Center for Neural Engineering, University of Minnesota, Minneapolis, United States

*For correspondence:
brett.foster@pennmedicine.
upenn.edu

**Abstract** Posterior cingulate cortex (PCC) is an enigmatic region implicated in psychiatric and neurological disease, yet its role in cognition remains unclear. Human studies link PCC to episodic memory and default mode network (DMN), while findings from the non-human primate emphasize executive processes more associated with the cognitive control network (CCN) in humans. We hypothesized this difference reflects an important functional division between dorsal (executive) and ventral (episodic) PCC. To test this, we utilized human intracranial recordings of population and single unit activity targeting dorsal PCC during an alternated executive/episodic processing task. Dorsal PCC population responses were significantly enhanced for executive, compared to episodic, task conditions, consistent with the CCN. Single unit recordings, however, revealed four distinct functional types with unique executive (CCN) or episodic (DMN) response profiles. Our findings provide critical electrophysiological data from human PCC, bridging incongruent views within and across species, furthering our understanding of PCC function.

## Editor's evaluation

This is an exciting manuscript that provides fundamental new insights into one of the most enigmatic brain regions; the posterior cingulate cortex. Using electrophysiological recordings from dorsal and ventral PCC subregions, the authors provide compelling evidence for a dorsal-executive and ventral-episodic functional subdivision. This paper will be of high interest to a broad range of neuroscientists.

## Introduction

Despite being routinely observed in studies of human brain function, the posterior cingulate cortex (PCC) remains one of the least understood neocortical areas, with little consensus as to its role in cognition (*Leech and Sharp, 2014*; *Leech and Smallwood, 2019*). Anatomically, PCC reflects the caudal aspect of the cingulate cortex, wrapping around the posterior trunk and splenium of the corpus callosum (*Vogt et al., 1995*; *Vogt et al., 2001*; *Margulies et al., 2009*; *Figure 1B*). Physiologically, PCC displays high basal levels of metabolic activity (*Raichle et al., 2001*; *Buckner et al., 2008*). Clinically, a growing literature suggests PCC is implicated in the progression of neurodegenerative and psychiatric disease (*Leech and Sharp, 2014*; *Vogt, 2019*). However, the deep medial location and rich vascularization of PCC (*Bernier et al., 2018*) limit the occurrence of focal insult and corresponding neuropsychological findings, often foundational to understanding many other brain regions (*Vaidya et al., 2019*).

At present, theorizing on the function of human PCC is typically limited to its role as a core node of the default mode network (DMN) (*Raichle et al., 2001*; *Fox et al., 2005*; *Fransson, 2005*; *Fransson and Marrelec, 2008*; *Buckner and DiNicola, 2019*). Whereby, a large functional neuroimaging literature shows that PCC, as part of the DMN, displays high basal levels of activity that are reduced during a variety of tasks, especially those requiring focused attention and cognitive control (*Shulman et al., 1997*; *Raichle et al., 2001*; *Harrison et al., 2008*; *Anticevic et al., 2012*). Consequently, the DMN is often reported as being anti-correlated or oppositional to other brain systems, such as the dorsal attention or cognitive control networks (DAN; CCN) that are engaged during attention demanding tasks (*Fox et al., 2005*; *Fransson, 2005*; *Uddin et al., 2009*). These findings led to the DMN, and in turn PCC, to be termed 'task-negative', as opposed to 'task-positive' regions (*Fox et al., 2005*; *Uddin et al., 2019*).

However, hemodynamic activity in PCC is not decreased in all tasks (*Spreng, 2012*). In particular, it increases during episodic memory retrieval (*Shannon and Buckner, 2004*; *Wagner et al., 2005*; *Vincent et al., 2006*) and other internally focused cognitive tasks associated with self-referential thought and envisioning past or future scenarios (*Svoboda et al., 2006*; *Hassabis and Maguire, 2007*; *Harrison et al., 2008*; *Spreng et al., 2009*; *Buckner and DiNicola, 2019*). These findings highlight an alternate dichotomy where PCC/DMN function is associated with 'internally', versus 'externally', focused cognition (*Buckner and Carroll, 2007*; *Andrews-Hanna et al., 2014*). Supporting these neuroimaging findings, human intracranial recordings of PCC and surrounding posteromedial cortex (PMC) suggest clear electrophysiological evidence of activity suppression during external attention (*Miller et al., 2009*; *Dastjerdi et al., 2011*; *Ossandón et al., 2011*; *Foster et al., 2012*; *Foster et al., 2015*) and activity enhancement during cued episodic thought (*Foster et al., 2012*; *Foster et al., 2015*; *Fox et al., 2018*).

In contrast, electrophysiological studies in the non-human primate have focused on the contributions of PCC to cognitive control (*Hayden et al., 2010*), decision making (*McCoy et al., 2003*; *McCoy and Platt, 2005*; *Pearson et al., 2009*), and value judgement (*Hayden et al., 2008*; *Heilbronner et al., 2011*) – conditions thought to anti-correlate with PCC engagement in humans. Together, single unit recordings in the macaque PCC during economic decisions have led to a hypothesized role in strategy selection (*Pearson et al., 2009*; *Pearson et al., 2011*; *Heilbronner and Platt, 2013*), further emphasizing a much more executive function, in apparent contrast to the common focus of human data. However, studies of human decision making and value judgements consistently observe PCC engagement (*Bartra et al., 2013*; *Clithero and Rangel, 2014*; *Oldham et al., 2018*), yet such findings have received limited attention (compared to anterior cingulate cortex) and therefore limited integration with studies of the DMN (*Acikalin et al., 2017*).

We hypothesized that these apparently contradictory findings reflect, in part, a failure to appreciate the specific anatomical organization of PCC. Contemporary studies of human brain network organization suggest that PCC contains putative subregions associated with multiple distinct functional systems (*Andrews-Hanna et al., 2010*; *Yeo et al., 2011*; *Leech et al., 2012*; *Bzdok et al., 2015*; *Glasser et al., 2016*; *Braga and Buckner, 2017*; *Gordon et al., 2017b*). For example, functional connectivity patterns are consistent with estimated structural connectivity, with prefrontal cingulum tracts entering primarily via the dorsal aspect of PCC, whereas medial temporal lobe tracts enter via its ventral aspect (*Greicius et al., 2009*). Evidence for PCC subregional organization also derives from homologous cytoarchitectural and connectivity data in humans and non-human primates (*Vogt et al.,*

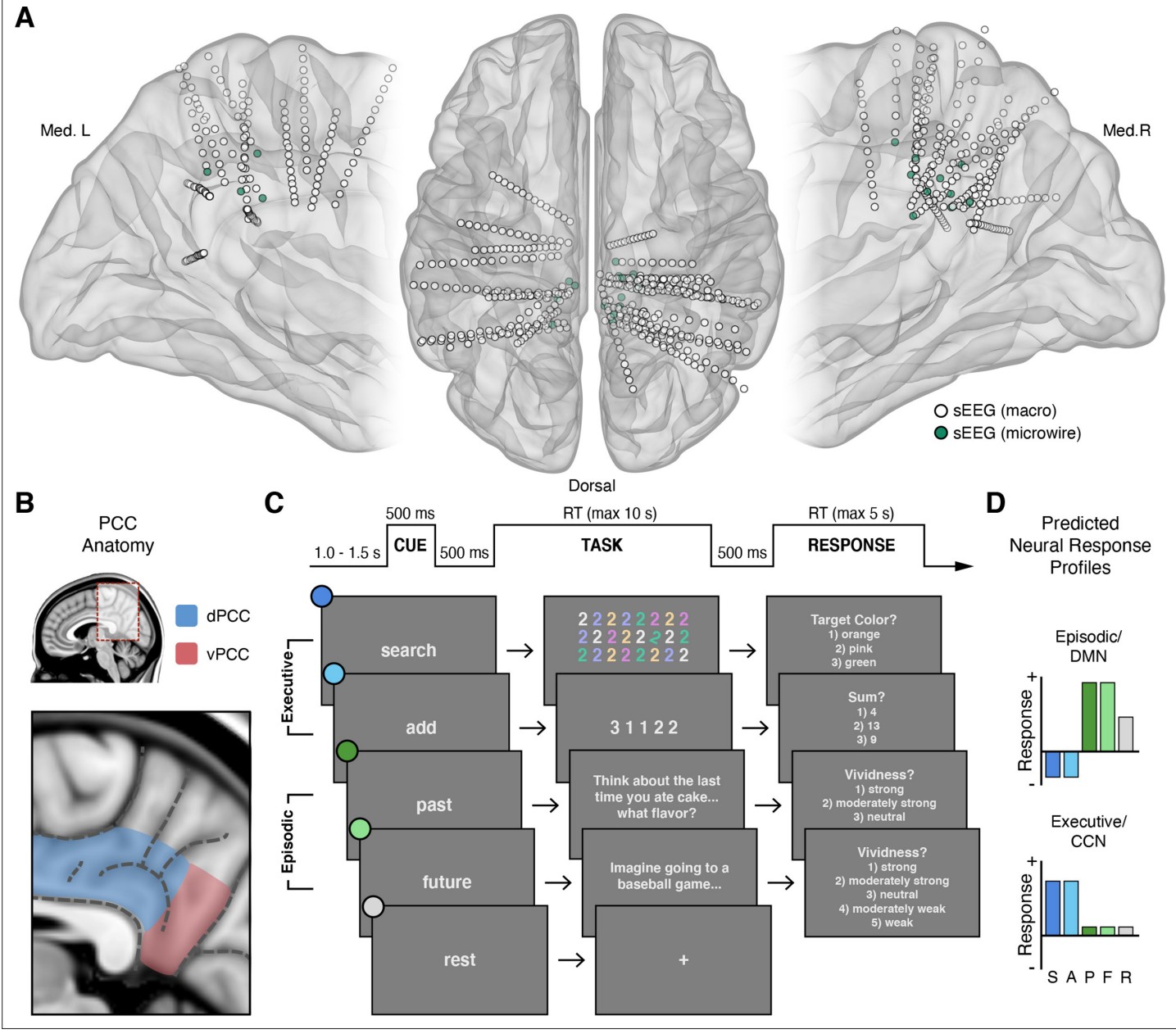

**Figure 1.** Intracranial recordings sites, PCC anatomy, task design, and predicted neural responses. (**A**) Anatomical locations, normalized to MNI space, of intracranial sEEG depth electrodes targeting the posterior aspect of the cingulate cortex, with the distal contact of probes containing microwires indicated in green. (**B**) Anatomy of human PCC, differentiating dorsal (dPCC) and ventral (vPCC) subregions and demarcating regional sulci following the most recent definitions in PCC (**Petrides, 2018**; **Willbrand et al., 2022**). (**C**) Attention/Memory Switch task procedure and example stimuli. The task included two executive attention conditions: search and add, two episodic memory conditions: past and future, and a rest condition. Each trial was proceeded by a cue indicating the forthcoming task condition and was followed by a trial response period, excluding rest (see Methods). (**D**) Schematic of the putative neural response profiles to the task conditions for episodic/DMN (default mode network) selectivity vs. executive/CCN (cognitive control network) selectivity.

*1995*; *Vogt et al., 2001*; *Vogt and Laureys, 2005*; *Parvizi et al., 2006*; *Vogt et al., 2006*; *Margulies et al., 2009*). Together, these data convey a dissociation of the dorsal and ventral PCC (*Figure 1B*). Such a division suggests that dorsal PCC (dPCC) may be associated with executive control systems of the frontal lobe and ventral PCC (vPCC) may be associated with episodic memory systems of the medial-temporal lobe (*Vincent et al., 2008*; *Spreng et al., 2010*; *Leech et al., 2011*). This subregional

perspective better reconciles functional views of PCC across species, as studies in the non-human primate have exclusively focused on the dPCC (often termed area *CGp*; *McCoy et al., 2003*).

As noted above, extant data from humans suggests executive attention demanding tasks typically suppress electrophysiological activity within human PCC, consistent with DMN function (*Buckner and DiNicola, 2019*). In contrast, electrophysiological evidence of PCC engagement during executive tasks, consistent with CCN function, is more limited (*Daitch and Parvizi, 2018*). Based on human neuroimaging and non-human primate electrophysiology, we hypothesized that subregions of PCC, specifically within its dorsal aspect, should be engaged during executive attention demanding tasks more than memory demanding tasks. Whereby, dPCC should reflect an electrophysiological response profile more consistent with the CCN rather than the DMN (*Figure 1D*).

To test this, we leveraged human intracranial electrophysiology via local field potential (LFP) and single neuron recordings targeting dPCC to capture neural activity during randomly alternating executive attention and episodic memory demanding tasks. We observed that neural populations within dPCC, as captured by the LFP, showed significantly greater responses (in broadband gamma, BBG; 70–150 Hz) during executive compared to episodic conditions. Strikingly, functional clustering of single unit activity revealed multiple functional cell types within dPCC. A majority of single units responded selectively to only one of the two executive attention tasks (visual search or arithmetic), suggesting distinct populations underlie the observed CCN-like LFP response. However, a third functional cell type showed clear selectivity to both episodic memory conditions (past and future event scenarios) as well as suppression to both executive task conditions, highly consistent with DMN function. A final functional cell type displayed increased firing during cued rest periods, also a feature of the DMN. Together, these data provide clear electrophysiological evidence for executive processing within dPCC, consistent with the non-human primate, but also suggest a rich functional organization at the single unit level. These findings are discussed within the context of a growing literature examining the varied functional organization of associative cortices, like PCC, via precision neuroimaging. Careful consideration of this rich functional organization will be critical for better understanding the role of PCC in cognition and disease.

## Results

### Intracranial PCC recordings during an attention/memory switch task

Electrophysiological activity in PCC was recorded by stereo-electroencephalography (sEEG) depth electrodes in 20 subjects undergoing invasive monitoring for epilepsy (female = 8; male = 12; see Methods). Across subjects, 35 electrodes were localized to the PCC region, 29 within dPCC and 6 within vPCC (*Figure 1A and B*). As the focus of this study, subsequent analyses concentrated on dPCC recording sites. Subjects performed an attention/memory switch task consisting of randomly alternating trials of executive attention conditions (search or add) and episodic construction conditions (past or future), engaging executive and episodic processes respectively (*Figure 1C*; see Methods). These conditions reflect tasks well known to differentially drive both CCN and DMN systems (*Figure 1D*) as observed via human neuroimaging and intracranial electrophysiology (*Jerbi et al., 2010*; *Ossandón et al., 2011*; *Foster et al., 2012*; *Ossandón et al., 2012*; *Foster et al., 2015*; *Daitch and Parvizi, 2018*; *Fox et al., 2018*). Task performance showed high accuracy for the executive conditions (mean accuracy: 86.47% for search; 92.59% for add) and robust vividness scores for the cued episodic conditions (modal vividness response: 'strong' for past; 'strong' for future).

### Task selectivity of LFP in dorsal PCC

For macro-electrode sEEG recordings, activation of local neural populations can be captured by amplitude changes in the broadband gamma (BBG) frequency range (70–150 Hz) of the recorded local field potential (LFP) (*Ray and Maunsell, 2011*; *Bartoli et al., 2019*). BBG activity is strongly correlated with both blood-oxygen-level-dependent (BOLD) fMRI and population firing rate, making it a useful marker for reconciling human neuroimaging and non-human primate electrophysiology findings (*Miller, 2010*). We found that dPCC displayed increased activity specifically within the BBG range during the cue and early task execution of the executive attention conditions, search and add. An example dPCC recording site and time-frequency response is shown in *Figure 2A and B*. Group data ($n_{probes}$ = 29, $n_{subjects}$ = 19) shows BBG activation within dPCC to be transient and peaked following

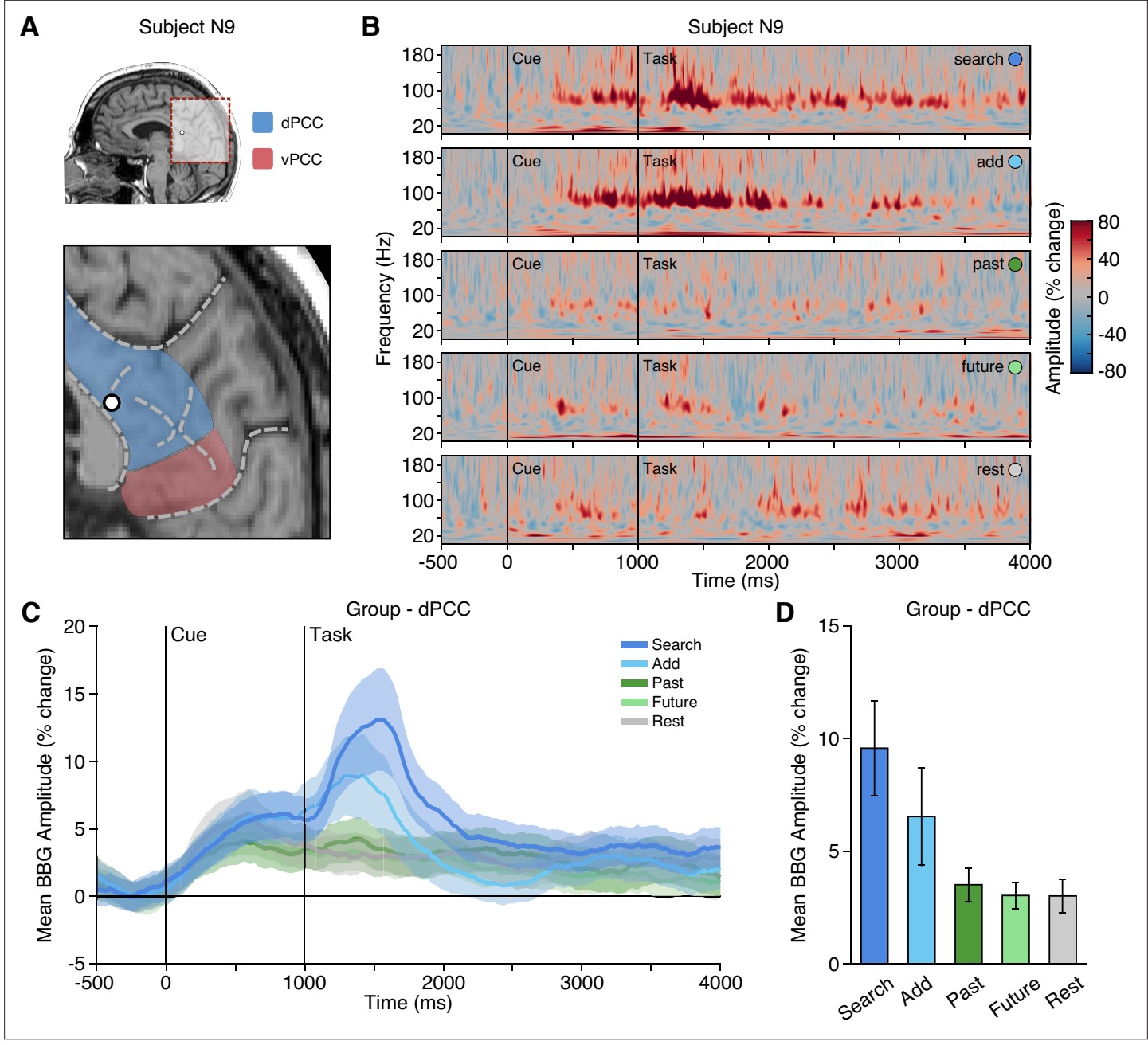

**Figure 2.** Executive condition selectivity of LFP BBG task responses in dPCC. (**A**) Location of a representative dPCC electrode in subject N9. (**B**) Time-frequency spectrogram for electrode in (**A**) depicting amplitude (% change) dynamics over time for each task condition. Amplitude in the BBG frequency range is increased during the cue and early task performance of attention conditions (search/add). (**C**) Group mean ($n_{probes}$ = 29, $n_{subjects}$ = 19) BBG amplitude (with s.e.m. shading) over time, aligned to trial onset, across conditions for all dPCC electrodes. Black vertical lines indicate time of cue and task onset. (**D**) Group mean BBG amplitude (with s.e.m.) during early task execution (task onset to 1000 ms) across conditions for all dPCC electrodes. Together, these data show increased activity of dPCC populations during early task execution of executive attention conditions, consistent with the CCN. See *Figure 2—figure supplement 1* for vPCC data.

The online version of this article includes the following figure supplement(s) for figure 2:

**Figure supplement 1.** Episodic condition selectivity of LFP BBG task responses in vPCC.

the onset of search and add tasks (*Figure 2C*). During early task execution, from task onset to 1000 ms (1000–2000 ms post-cue), group mean BBG amplitude was greatest for the search condition, followed by the add condition, with low mean responses for past, future, and rest conditions. A mixed-effects model with task as a fixed effect and subject/electrode as random effects, revealed a significant main

effect of task on BBG amplitude for the same task window. Specifically, mean BBG amplitude was significantly higher for the search task compared to add, past, future, and rest conditions (search-add: t(5093)=3.859, p=0.0011; search-past: t(5093)=8.092, p<0.0001; search-future: t(5093)=8.573, p<0.0001; search-rest: t(5093)=8.706, p<0.0001), and furthermore, was significantly higher for the add task compared to past, future, and rest conditions (add-past: t(5093)=4.236, p=0.0002; add-future: t(5093)=4.717, p<0.0001; add-rest: t(5093)=4.847, p<0.0001). These data provide clear support for an executive functional response profile within dPCC, which is highly consistent with the CCN and oppositional to that predicted for the DMN (*Figure 1D*). In contrast, sites within vPCC ($n_{probes}$ = 6) showed the converse BBG response profile across conditions, consistent with the DMN and prior observations (*Dastjerdi et al., 2011*; *Foster et al., 2012*; *Foster et al., 2015*), where BBG responses were greater during late task execution (2000–3000 ms after task onset) for the episodic task conditions (see *Figure 2—figure supplement 1*). A mixed effects model revealed a significant main effect of condition on BBG amplitude, with vPCC mean BBG amplitude being significantly higher for past than all other conditions (past-add: t(1253)=8.841, p<0.0001; past-search: t(1253)=5.151, p<0.0001; past-rest: t(1253)=3.978, p=0.0007; past-future: t(1253)=3.020, p=0.0216), and significantly lower for add than all other conditions (add-past: t(1253)=–8.841, p<0.0001; add-future: t(1253)=–5.811, p<0.0001; add-rest: t(1253)=–4.863, p<0.0001; add-search: t(1253)=–3.690, p=0.0022).

## Anatomical organization of PCC activity

To examine the distribution of responses across PCC recording sites, all electrode locations were normalized to MNI space and depicted with their respective mean BBG amplitude for early and late time periods of task execution for the combined executive (search/add) and episodic (past/future) conditions (*Figure 3*; See Methods). As shown in *Figure 3A*, executive activity during an early time window (task onset to 1 s) was maximal for sites closer to the callosal sulcus within dPCC (i.e. inferior aspect of dPCC). When considering a later time period of task execution, 2–3 s after task onset, population selectivity appeared to follow a more dorsal/ventral gradient with executive activity more dorsal and mnemonic activity more ventral within PCC (*Figure 3C/D*). While PCC is often viewed as synonymous with DMN function, human neuroimaging often reports on executive control tasks engaging dPCC (*Leech et al., 2011*). Indeed, meta-analytic association of neuroimaging studies for the term 'cognitive control' identifies commonly reported activations within the inferior aspect of dPCC (Neurosynth association test; n studies = 598). The centroid of this cluster (threshold z>4) is shown in *Figure 3* (green hexagon), proximal to many recording sites.

## Task selectivity of single units in dorsal PCC

Microwires extending from the distal tip of a subset of sEEG probes allowed for recording, detecting, and sorting single unit spiking activity within dPCC ($n_{probes}$ = 6; $n_{units}$ = 91; See Methods). Overall, PCC neurons varied in their mean firing rate, but predominately displayed sparse firing properties, with a group mean spike rate of 1.7 Hz (across entire task). Initially, when averaging all units, group mean firing rates suggested limited task specific responses and wide variability (*Figure 4—figure supplement 1A*). Inspection of individual firing rates indicated this was due to a diversity of opposing functional selectivity profiles across units. Therefore, we applied unsupervised hierarchical clustering of units based on the similarity of task response profiles to identify putative functional types (See Methods; *Figure 4—figure supplement 1*). Strikingly, this clustering revealed four clear functional types, with several unique response features. *Figure 4* shows example unit raster plots and instantaneous firing rates over time for each of the four identified cell types. Two functional types (1 and 2) showed increased firing rates to the executive task conditions, consistent with LFP findings. However, these units displayed a more specific level of selectivity than that of the population data. These two functional types showed condition-specific selectivity within the executive tasks, where Type 1 ($n_{units}$ = 13) was selective for the search condition only, whereas Type 2 ($n_{units}$ = 34) was selective for the add condition only. Similar to LFP BBG responses, Type 1 and 2 units showed increased activity beginning during the cue period and continuing to increase with task engagement. Overall, Type 1 and 2 unit groups are consistent with LFP evidence for CCN-like executive processing within dPCC but suggest distinct functional cell types underlie the observed LFP BBG responses. In contrast to the LFP data, two additional functional unit groups reliably showed response profiles consistent with the DMN. Whereby, Type 3 units ($n_{units}$ = 26), displayed firing rates that were increased for both the past and

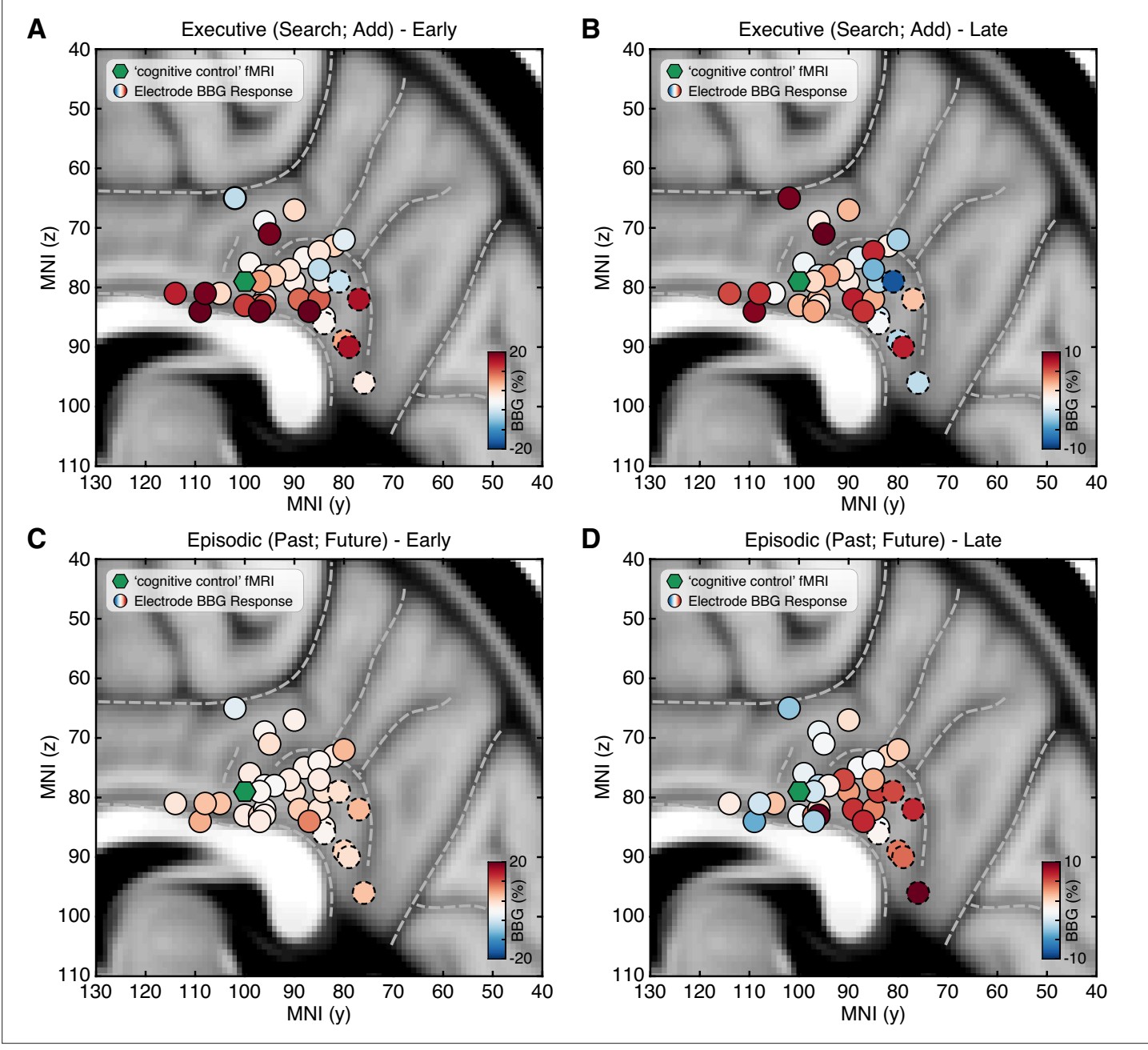

**Figure 3.** Spatial clustering of LFP BBG responses across PCC. Mean LFP BBG response for each macro-electrode site within PCC averaged for the executive (search/add) (A/B) and episodic (past/future) (C/D) task conditions during early (task onset to 1000 ms; left) and late (2000–3000 ms into task; right) task execution. Electrode coordinates are shown in MNI voxel space, collapsed for the x-dimension (i.e. collapsed for left/right hemispheres). Green symbol indicates the centroid maxima of the association map for search term 'cognitive control' at z>4.0 from Neurosynth meta-analysis. BBG responses predominate for the executive task conditions during the early task window, specifically for electrodes more proximal to the callosal sulcus. However, BBG responses to the episodic task conditions emerge during the late task window, specifically for more ventral sites. Ventral PCC electrodes (n=6) are indicated with a dashed outline.

future episodic conditions, weakly elevated for the rest condition, and suppressed for both executive task conditions (search/add; *Figure 4C*). Such a response profile is highly consistent with the DMN predicted responses (*Figure 1D*). Furthermore, unlike Type 1 and 2 units, Type 3 units showed a more delayed increase in firing rate, occurring well after the start of task engagement. Finally, Type 4 units ($n_{units}$ = 18) primarily displayed increased firing rates to the rest condition, which is another related functional property often associated with the DMN (*Dastjerdi et al., 2011*). These units, which

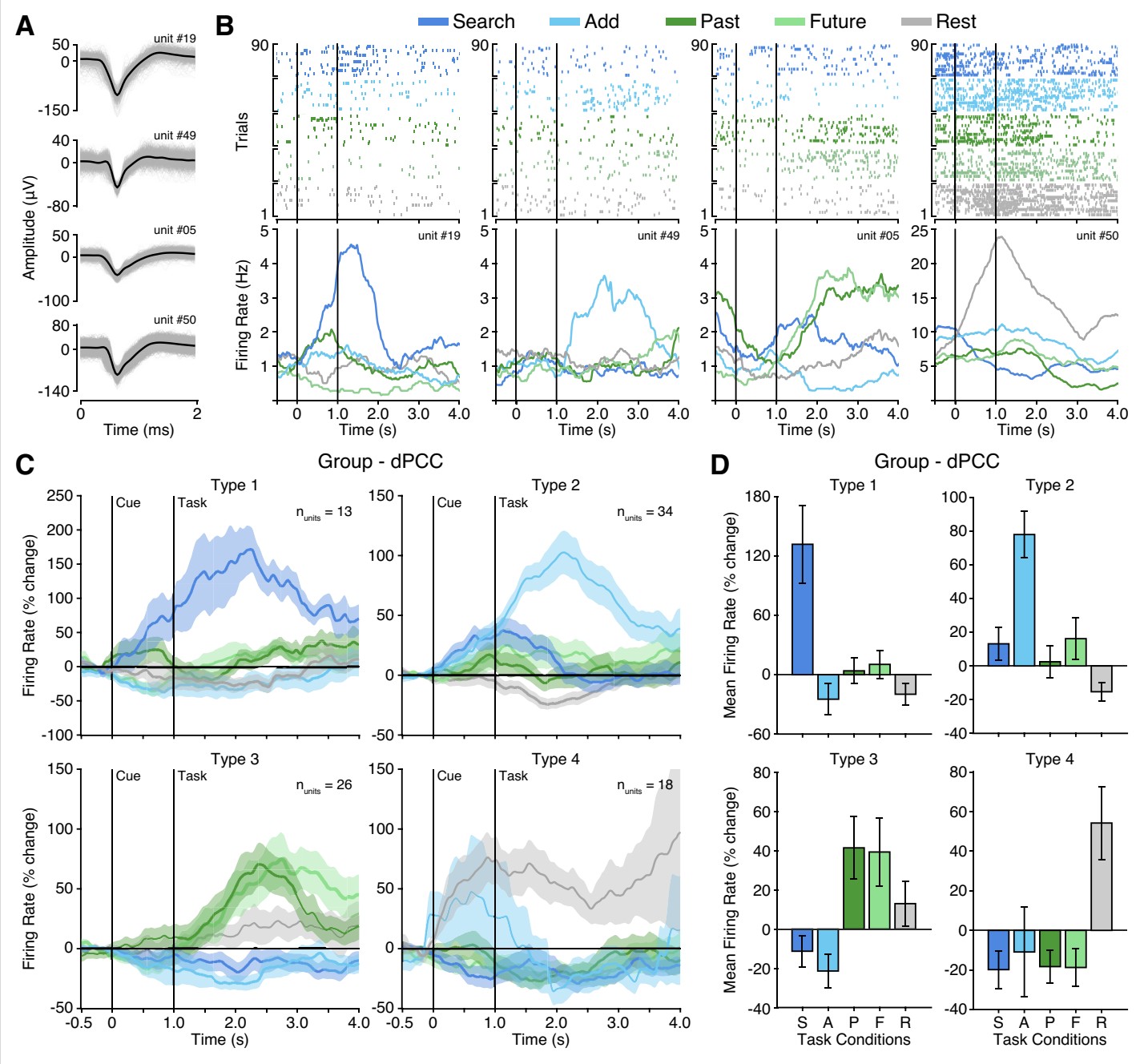

**Figure 4.** Four functional single unit types in dPCC. (**A**) Example isolated single unit waveforms. (**B**) Raster plots and instantaneous firing rate traces for four example isolated single units shown in (**A**). Example units show distinct functional selectivity profiles across task conditions. (**C**) Mean instantaneous firing rates (with s.e.m. shading) across task conditions for the four functional single unit types identified based on unsupervised similarity clustering (See Methods & *Figure 4—figure supplement 1*; Type 1 $n_{units}$ = 13; Type 2 $n_{units}$ = 34; Type 3 $n_{units}$ = 26; Type 4 $n_{units}$ = 18). (**D**) Mean firing rate (with s.e.m.) for the four functional single unit types (averaged over task onset to 2 s into task execution). In (**B**) and (**C**), black vertical lines indicate time of cue and task onset.

The online version of this article includes the following figure supplement(s) for figure 4:

**Figure supplement 1.** Single unit response clustering.

typically showed higher basal firing rates, increased firing after the cue was presented. Consistent with our observations, there was initially no significant main effect of task condition on firing rate (p=0.55) prior to considering cluster type. There was, however, a significant effect of cluster type on mean task firing rate (p=0.01) and a significant interaction between task condition and cluster type (p<0.0001).

To quantify the consistency of cluster membership, we performed hierarchical clustering through a complete leave-one-out process, resulting in a 94% identical cluster rate. Importantly, each of the four functional cell types were observed on multiple microwire probes, indicating that none of the cluster types originated from a single subject or recording site. Furthermore, each microwire probe gave rise to units of different selectivity types, suggesting a more mixed organization at the single unit level. Together, these findings in part support LFP observations, but further suggest a richer and more complex array of functional organization within dPCC. These data provide the first single unit evidence of both CCN and DMN function within human PCC.

## Discussion

Intracranial recordings targeting human dPCC revealed strong electrophysiological evidence of executive task engagement. Dorsal PCC neural populations, as captured by LFP broadband gamma activity, displayed executive processing activity profiles consistent with that of cognitive control regions, as opposed to the episodic/DMN activity commonly attributed to the entire PCC (*Fox et al., 2018*). Across-subject consideration of the spatial organization of population responses showed some clustering of executive selectivity inferior within dPCC near the callosal sulcus. The unique opportunity to record single neuron activity in human dPCC illuminated a far richer functional heterogeneity, beyond what could be discerned at the population level. Neurons in dPCC exhibited distinct selectivity profiles for specific executive attention tasks (search or add), consistent with, but more selective than, LFP data. In addition, while not observed in the LFP, a distinct set of neurons showed a clear DMN type of functional profile selective to either episodic processing (past and future) conditions or rest periods. Together, these findings allow for reconciliation of conflicting reports of PCC function between species and begin to unveil the complex functional contributions of single neurons in dPCC to executive and episodic processing.

### Executive processing in human PCC

Historically, human PCC has been functionally considered within its role as a member of the DMN, active during internally focused cognition and memory retrieval and deactivated during attention demanding tasks (*Buckner et al., 2008*; *Buckner and DiNicola, 2019*). Such a functional role of PCC is in direct contrast to the functional view predominating from non-human primate electrophysiology studies, examining its contributions to executive processes including strategy selection and decision making (*Pearson et al., 2011*). Based on the common dorsal targeting of PCC recordings in these studies (termed area *CGp*), we sought to reconcile conflicting reports by focusing on the electrophysiology of human dPCC during a task that required performance of both executive and episodic processing. We found that neural populations in human dPCC showed significantly stronger BBG amplitude LFP responses during executive attention conditions (search/add) than episodic retrieval conditions, reflective of executive processing. Thus, when specifically examining dorsal PCC, population selectivity is more cohesive with the non-human primate electrophysiological data than the traditional DMN view of PCC in the human literature. By providing electrophysiological evidence of executive activity within human dPCC, these findings provide evidence of homology between the functions of macaque area CGp and human dPCC (i.e. area 23d in both species *Vogt et al., 1987*; *Vogt et al., 1995*). Further, they highlight the importance of considering subregional variation in PCC. Importantly, these findings are consistent with a selection of literature from human neuroimaging, typically unincorporated with the DMN/memory literature (*Acikalin et al., 2017*), which implicates specifically dPCC with executive control processes as well as certain aspects of value-based decisions (*Bartra et al., 2013*; *Clithero and Rangel, 2014*; *Oldham et al., 2018*).

### Functional organization of PCC

The heterogeneity of functional responses in PCC prompts the question of the spatial organization of the region. As previously noted, cytoarchitecture, structural connectivity, and network parcellations suggest subregional distinctions between dorsal and ventral PCC (*Vogt et al., 1995*; *Vincent et al., 2008*; *Greicius et al., 2009*). Electrophysiology studies, though limited, have alternatively suggested a more mixed functional mosaic throughout PCC and posteromedial cortex more broadly (*Daitch and Parvizi, 2018*). Our findings reveal clustering of early executive selectivity inferiorly within

dPCC along the callosal sulcus, adding to evidence for a dorsal/ventral gradient for executive and episodic responses. Electrodes with executive activity were observed proximal to voxels associated with 'cognitive control' in fMRI studies, linking our findings with dPCC activations appearing in neuroimaging data. These results begin to illuminate that there is some functional organization of heterogeneous activity on the population level, as opposed to a totally mixed mosaic. Population selectivity appears to vary by dorsal/ventral subregions, though the division between subregions is complex and likely subject to individual differences. Regarding the present study, we note that our focus was on dPCC, with vPCC being more sparsely sampled, limiting a full characterization of functional gradients. However, we also note that our vPCC findings are consistent with a number of intracranial studies showing DMN-like responses within PCC (*Fox et al., 2018*). Precision fMRI within individuals emphasizes the impact of individual variability, particularly in associative cortices and their putative networks (*Braga and Buckner, 2017*). Indeed, precision neuroimaging specifically highlights the dPCC as region of interest in assessing individual variability (*Gordon et al., 2017a*). A much larger spatial sampling of recording sites throughout dorsal and ventral PCC, in addition to further consideration of individual anatomy, will be required to determine this organization. As precision functional data might not always be available for individuals, how might functional organization within PCC be inferred? Our electrophysiological observations are broadly consistent with recent large-scale examination of PCC sulcal anatomy which identified a new tertiary sulcus (inframarginal sulcus) as a useful landmark for the transition from DMN to CCN within dPCC (*Willbrand et al., 2022*). Further integration of invasive electrophysiology and precision neuroimaging within subjects will be critical for elucidating key features of PCC functional neuroanatomy and its variability across individuals (*Gordon et al., 2017b*).

## Single neuron executive and episodic selectivity

Acquisition of single neuron spiking activity within human dPCC unveiled distinct functional profiles and organizational principles not observable through population recordings alone. Functional clustering of single unit responses during the attention/memory switch task revealed four unique functional cell types within dPCC. Interestingly, the majority of dPCC neurons displayed specific selectivity for different executive tasks, preferentially responding to one of the two attention conditions (search or add). This condition-specific differentiation is not apparent in the population data, where LFP responses were typically seen for both or neither of the attention conditions. A third type of dPCC neuron possessed episodic selectivity, with increased firing rates for both the past and future conditions, yet suppression for both executive conditions. This indicates that episodic/DMN activity, the focus of human neuroimaging work, does indeed exist in dPCC, though appears to be outweighed by executive activity when looking at the population level. Finally, a fourth type included neurons with high rest activity, fitting the original historical view of the DMN as task-negative, suggesting that the high basal activity of PCC during rest is potentially driven by a subset of neurons differentiable from those active during retrieval (*Dastjerdi et al., 2011*). Notably, differences between the LFP and single unit data result in varying interpretations of PCC organization. Each microwire probe gave rise to units of multiple functional selectivity types, revealing a more mixed organization on the single unit level. This considerable heterogeneity at the neuronal level likely contributes to the complex pattern of spatially overlapping yet functionally distinct subregions within dPCC, reflecting a convergence of multiple brain networks, as observed by data-driven functional connectivity maps within the region (*Leech et al., 2011*; *Leech et al., 2012*). Our findings provide the first single unit evidence of executive and episodic function within human PCC, and in turn the CCN and DMN. Further single-unit studies during cognitive control and decision-making tasks will be vital for more precise identification of PCC neural selectivity during executive behavior and improved understanding of the functional homology with the non-human primate.

In conclusion, our findings bridge previously incongruent views of PCC function between fields by targeting the dPCC subregion and identifying executive processing in human PCC populations. Further, this work represents the first human single neuron evidence of executive and episodic selectivity in PCC, revealing the complex functional contributions of dPCC neurons. Together, they inform a view of PCC as a heterogeneous region composed of dorsal and ventral subregions specializing in executive and episodic processing respectively. Careful consideration of this functional neuroanatomy and its further elucidation will be critical for better understanding the role of PCC in cognition and disease.

## Methods

### Human subjects

Intracranial recordings from the human posterior cingulate cortex were obtained from 20 subjects (N1-N20; 8 female, 12 male; mean age 34.1 years., range 19–56 years) undergoing invasive monitoring as part of their treatment for refractory epilepsy at Baylor St. Luke's Medical Center (Houston, Texas, USA). Subject information is detailed in *Supplementary file 1*. All decisions regarding sEEG trajectories were made solely on the basis of clinical criteria. Experimental protocols were approved by the Institution Review Board at Baylor College of Medicine (IRB protocol number H-18112), with subjects providing verbal and written informed consent to participate in this study. We did not include patients with epileptic foci, anatomical abnormalities, or prior surgical resection in posteromedial regions.

### Electrode arrays

Recordings were performed using sEEG depth electrodes (PMT Corp., MN, USA) and Behnke-Fried macro-micro probes (Ad-Tech Medical Instrument Corp., WI, USA). Clinical depth electrodes consist of 8–12 macro contacts, each 1.3–2.0 mm in diameter, along the electrode shaft. In addition to macro contacts, Behnke-Fried electrodes also included a bundle of 8 shielded microwires and 1 unshielded reference microwire, 38–40 μm in diameter each, extending approximately 4 mm from the distal tip. We recorded from 35 PCC electrodes across subjects, 14 of which contained microwires.

### Behavioral task

Subjects performed an attention/memory switch task consisting of randomly alternating trials of executive attention and episodic memory conditions (see *Figure 1C*). The task included two attentional conditions (search and add), two episodic conditions (past and future), and a rest condition. At the onset of each trial, subjects were presented with a cue indicating the upcoming condition (e.g., 'SEARCH') for 500 ms. Next, for each trial excluding rest conditions, the task screen was presented until subjects provided a button press to indicate task completion or a maximum of 10 s was reached (mean duration 7.27 s). The rest period lasted for a fixed 4 s duration. Finally, subjects were asked to provide a condition-specific answer during a response period. During the search condition, subjects were presented with a matrix of multi-colored letters or numbers and asked to find the one letter/number that was slightly rotated, then respond correctly by indicting the color of the target from five alternative options. During the add condition, subjects were asked to calculate the sum of five single digit numerals and then respond correctly by indicating the correct sum from five alterative options. Prior to testing, subjects were instructed to perform the search and add conditions as quickly and accurately as possible. During the episodic conditions (past and future), subjects were presented with statements that either prompted them to recall a past event (e.g. 'Think about the last time you ate cake. What flavor was it?') or to imagine a future event (e.g. 'Imagine going to a baseball game.'). Prior to testing, subjects were instructed to visualize specific past or future events in as much detail as possible. Subjects were required to rate their recall/imaginary experience on a vividness scale of five options: 'strong', 'moderately strong', 'neutral', 'moderately weak', or 'weak'. During the rest condition, subjects fixated on a central cross for 4 s with no response period. There were 18 trials per condition, for a total of 90 trials per task run. Subjects typically performed 2 unique runs of the task, each approximately 15–20 min in duration. All data analyses combine block runs. The task was presented on an adjustable monitor (1920 x 1,080 resolution, 47.5 x 26.7 cm screen size) at a viewing distance of 57 cm. Psychtoolbox functions (v3.0.12) running in MATLAB (v2017a, MathWorks, MA, USA) were used to program the experiment. A total of 40 blocks of the task were completed across the 20 subjects.

### Electrophysiological recording

Data was acquired at two electrophysiological scales: macroelectrode LFP and microwire single unit recordings. LFP data was acquired at a sampling rate of 2 kHz and bandpass of 0.3–500 Hz using a Blackrock Cerebus (Blackrock Microsystems, UT, USA) system. Initial recordings were referenced to a selected depth electrode contact within the white matter, distant from grey matter or pathological zones. Single unit data was acquired at a sampling rate of 30 kHz using either a Blackrock Cerebus ($n_{subjects}$ = 13; Blackrock Microsystems, UT, USA) system or Ripple Micro2 ($n_{subjects}$ = 1; Ripple Neuro, UT,

USA). For each Behnke-Fried probe, the 8 microwires were internally referenced to a 9th unshielded microwire. During all recordings, stimulus presentation was tracked using a photodiode sensor (attached to stimulus monitor) synchronously recorded at 30 kHz. All additional data processing was performed offline.

### Electrode localization and selection

To identify electrodes located within the posterior cingulate cortex, a post-operative CT scan was co-registered to a pre-operative T1 anatomical MRI scan for each subject, using FSL and AFNI (*Cox, 1996*; *Jenkinson et al., 2012*). The volume location of each electrode was identified by clear hyper-intensities on the aligned CT using AFNI and visualized using iELVis software functions in MATLAB (v2016a, MathWorks, MA, USA) (*Groppe et al., 2017*). Electrodes located within Brodmann areas 23 and 31, with boundaries defined by the marginal ramus of the cingulate sulcus, corpus callosum, parieto-occipital sulcus, and splenial sulcus, were identified and included in this study. To compare electrode locations across subjects, identified electrode coordinates were transformed to the standard Montreal Neurological Institute (MNI) 152 space (*Evans et al., 2012*). Transformation of electrodes to MNI space was performed using the ANTs function antsRegistrationSyN.sh (*Avants et al., 2011*; *Tustison and Avants, 2013*). After first aligning each subject's anatomical T1 scan to the MNI template, the resulting transform parameters for each subject were then used to perform the same alignment of electrode coordinates.

### Macroelectrode signal preprocessing

All macro sEEG electrode signal processing was performed using custom scripts in MATLAB (v2018b, MathWorks, MA, USA). First, raw signals were inspected for line noise, recording artifacts, or interictal epileptic spikes. Electrodes with clear epileptic or artifactual activity were excluded from further analysis. Next, all electrode contacts were notch filtered (60 Hz and harmonics). Following a bipolar referencing scheme, each macro channel was re-referenced to its adjacent channel on the same probe. Finally, all signals were down sampled to 1 kHz.

### Spectral decomposition of LFP activity

Filtered, re-referenced, and down sampled LFP signals were spectrally decomposed using Morlet wavelets, with center frequencies spaced linearly from 2 to 200 Hz in 1 Hz steps (7 cycles). Broadband gamma (BBG) was defined as neural activity between 70 and 150 Hz (*Bartoli et al., 2019*). The instantaneous amplitude was normalized into a percent change signal by applying a baseline correction at each time-frequency point (baseline division, using the average pre-stimulus values −500 ms to 0 ms for each trial). For group analysis, electrodes with multiple recording blocks were first averaged together for an electrode mean, after which electrodes were averaged together for a group mean.

### Microwire signal preprocessing

Microwire signals were bandpass filtered between 300 and 3000 Hz prior to spike extraction. Spike detection and sorting was done with Wave_clus 3 (*Chaure et al., 2018*) in MATLAB (v2018b, MathWorks, MA, USA). The voltage threshold used for spike detection was a negative threshold of the signal mean plus 5 standard deviations. Isolated waveforms were segregated in a semi-automated manner. The times of threshold crossing for identified units were retained for further analysis. Based on previously used criterion (*Rey et al., 2020*), units were classified as single- or multi-units based on the spike shape and ISI distribution of the cluster (single-units with <1% of spikes with an ISI less than 3ms). Across the 14 Behnke-Fried probes implanted, 8 were excluded due to noise or failure to detect single unit activity, which included the single vPCC microwire probe. Across the remaining 6 Behnke-Fried probes, 95 single-units were isolated.

### Functional grouping of single unit activity

To examine firing rate properties of single unit activity, spike times were convolved with a 10 ms (standard deviation) Gaussian kernel to create an instantaneous firing rate. The distribution of firing rates was then inspected, with three outlier units (>5 standard deviations) and one duplicate unit identified and removed. Averaging the event-related instantaneous firing rate of the remaining 91 units produced limited evidence of any robust condition responses and displayed wide variance

(*Figure 4—figure supplement 1A*). Subsequent inspection of the individual unit event-related activity revealed a diversity of functional response profiles and differing basal firing rates (effectively cancelling condition specific effects when averaged; see *Figure 4—figure supplement 1*). We therefore normalized basal firing rate differences by converting instantaneous firing rate from Hz to percent change relative to the pre-cue period, as done for the LFP data. Next, we concatenated the mean instantaneous firing rate (% change) for each task condition (from cue to 3 s post-cue) to create a task response feature vector for each unit (i.e. for each unit a time series was constructed by concatenating the mean responses over the first 3 s of each condition). This vector captures both condition selectivity as well as response time course for each unit. To identify any putative similarity between the functional responses of units, we performed unsupervised hierarchical clustering, as follows. First, all concatenated unit task response vectors noted above were correlated with each other to create a correlation or similarity matrix. Next, correlation values were inverted (1 – r) to obtain a distance matrix that was then subject to agglomerative hierarchical clustering (via *linkage* function in MATLAB). The resulting cluster tree or dendrogram was then inspected and revealed four main cluster branches (based on 90% max distance cut-off; *Figure 4—figure supplement 1D*). Units were then grouped in these clusters to define functional Types 1–4. It is important to note, this unsupervised approach does not require specification of cluster number. To assess the consistency of cluster membership across units, we repeated the above clustering while iterating through a complete leave-one-out process (90 iterations). This resulted in a 94% (84/90) identical cluster rate. In addition, across the four clusters identified, units from multiple subjects were present in each (Type 1=4/6; Type 2=6/6; Type 3=5/6; Type 4=4/6). Therefore, the resulting clusters did not simply reflect the task or sample structure.

## Statistical analysis

Analysis of BBG was performed using mixed-effects models with subject and electrode as random effects to account for nesting of multiple electrodes within each subject. To test pairwise differences between task conditions, Tukey's method was used with p-values adjusted for comparing a family of 5 estimates. For single unit data, a two-way ANOVA was used to examine the effects of task condition and cluster type on mean firing rate, as well as the interaction between task and cluster type. Statistical analyses were carried out using R statistical software.

## Additional information

### Competing interests
Sameer A Sheth: has consulting agreements with Boston Scientific, Neuropace and Zimmer Biomet. The other authors declare that no competing interests exist.

### Funding

| Funder | Grant reference number | Author |
| --- | --- | --- |
| National Institute of Mental Health | R01MH129439 | Brett L Foster<br>Benjamin Y Hayden |
| National Institute of Mental Health | R01MH106700 | Sameer A Sheth |
| National Eye Institute | R01EY023336 | Daniel Yoshor |
| National Institute of Mental Health | R01MH116914 | Brett L Foster |
| Eunice Kennedy Shriver National Institute of Child Health and Human Development | R21HD100858 | Kevin S Weiner |
| National Science Foundation | CAREER Award 2042251 | Kevin S Weiner |

The funders had no role in study design, data collection and interpretation, or the decision to submit the work for publication.

## Author contributions

Lyndsey Aponik-Gremillion, Conceptualization, Data curation, Software, Formal analysis, Validation, Investigation, Visualization, Methodology, Writing – original draft, Writing – review and editing; Yvonne Y Chen, Hernan G Rey, Software, Formal analysis, Supervision, Investigation, Visualization, Writing – review and editing; Eleonora Bartoli, Software, Supervision, Investigation, Visualization, Methodology, Writing – review and editing; Seth R Koslov, Formal analysis, Visualization, Writing – review and editing; Kevin S Weiner, Supervision, Visualization, Writing – review and editing; Daniel Yoshor, Resources, Supervision, Methodology; Benjamin Y Hayden, Conceptualization, Supervision, Methodology, Writing – review and editing; Sameer A Sheth, Resources, Supervision, Investigation, Methodology; Brett L Foster, Conceptualization, Formal analysis, Supervision, Funding acquisition, Investigation, Visualization, Methodology, Writing – original draft, Project administration, Writing – review and editing

## Author ORCIDs

Yvonne Y Chen 
Eleonora Bartoli 
Seth R Koslov 
Hernan G Rey 
Kevin S Weiner 
Sameer A Sheth 
Brett L Foster 

## Ethics

Human subjects: Experimental protocols were approved by the Institution Review Board at Baylor College of Medicine (IRB protocol number H-18112), with subjects providing verbal and written informed consent to participate in this study and have subsequent findings published.

## Decision letter and Author response

Decision letter https://doi.org/10.7554/eLife.80722.sa1
Author response https://doi.org/10.7554/eLife.80722.sa2

---

# Additional files

## Supplementary files

• Supplementary file 1. Subject demographic and electrode information. Demographic and electrode information is reported for each subject (1-20), including sex (male/female), age at time of experiment (years), total number of PCC electrodes, number of dPCC electrodes, number of vPCC electrodes, and number of microwire probes.

• MDAR checklist

## Data availability

The datasets and custom code for the current study are accessible through the National Institute of Mental Health Data Archive (NDA) collection https://doi.org/10.15154/1527891.

The following dataset was generated:

| Author(s) | Year | Dataset title | Dataset URL | Database and Identifier |
|---|---|---|---|---|
| Foster BL | 2022 | Posterior cingulate cortex and executive control of episodic memory | https://dx.doi.org/10.15154/1527891 | NIMH Data Archive, 10.15154/1527891 |

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
