## [Editor Report]

This is an exciting manuscript that provides fundamental new insights into one of the most enigmatic brain regions; the posterior cingulate cortex. Using electrophysiological recordings from dorsal and ventral PCC subregions, the authors provide compelling evidence for a dorsal-executive and ventral-episodic functional subdivision. This paper will be of high interest to a broad range of neuroscientists.

---

## [Decision Letter]

**Decision letter after peer review:**

Thank you for submitting your article "Distinct population and single-neuron selectivity for executive and episodic processing in human dorsal posterior cingulate" for consideration by *eLife*. Your article has been reviewed by 3 peer reviewers, one of whom is a member of our Board of Reviewing Editors, and the evaluation has been overseen by Timothy Behrens as the Senior Editor. The following individuals involved in the review of your submission have agreed to reveal their identity: Robert Leech (Reviewer #2); Xaver Funk (Reviewer #3).

Essential revisions:

1. Why was only broadband γ considered? I think this is fine, but in terms of relating to other datasets (especially extra-cranial) does limit the interpretation somewhat.

2. With the clustering, I was slightly unsure how this was performed; it would be useful to have more detail in terms of how the similarity matrix was generated. Also, with the statistics, comparing tasks/clusters probably needs some cross-validation, since the clustering is performed on task features (I think) and will bias towards finding significant differences in the ANOVA.

3. What is the spatial distribution of the different clusters? Is this something that can be plotted? If so, it would be potentially very interesting.

4. The statistical analysis of BBG in the ventral electrodes should be reported and vPCC electrodes should be indicated in Figure 3.

5. Please include an analysis of the vPCC microwire probe.

6. In case the statistics and locations of vPCC electrodes do not support the above-mentioned conclusion, I recommend that it should be qualified accordingly to include a statement about the vPCC data not being completely conclusive and that further complementary research within this region is warranted.

7. Participants were excluded if they presented with frank anatomical abnormalities, however, it is not clear how this decision was made. Even if gross anatomical abnormalities are not evident, subtle changes in structural and functional connectivity might be present in this group. I wonder if the authors might therefore comment on how representative these results are to inform the functional architecture of the PCC and its subdivisions in general.

*Reviewer #1 (Recommendations for the authors):*

I thoroughly enjoyed reading this manuscript and believe it will make an important contribution to the literature on the DMN, as well as episodic memory more broadly.

My main query is in relation to the generalisability of the current findings given that these participants are undergoing invasive monitoring for refractory epilepsy. The authors mention that participants were excluded if they presented with frank anatomical abnormalities, however it is not clear how this decision was made. Even if gross anatomical abnormalities are not evident, subtle changes in structural and functional connectivity might be present in this group. I wonder if the authors might therefore comment on how representative these results are to inform the functional architecture of the PCC and its subdivisions in general.

*Reviewer #2 (Recommendations for the authors):*

I have some suggestions that might improve (mainly the presentation) of the manuscript.

Why was only broadband γ considered? I think this is fine, but in terms of relating to other datasets (especially extra-cranial) does limit the interpretation somewhat.

With the clustering, I was slightly unsure how this was performed: it would be useful to have more detail in terms of how the similarity matrix was generated. Also, with the statistics, comparing tasks/clusters probably needs some cross-validation, since the clustering is performed on task features (I think) and will bias towards finding significant differences in the ANOVA.

What is the spatial distribution of the different clusters? Is this something that can be plotted? If so, it would be potentially very interesting.

I am not clear if the data and code are going to be made openly available if the paper is accepted. Could this be clarified?

The results indicating considerable heterogeneity at a neuronal level might be interesting to interpret in the context of our paper (Leech et al., 2012, Journal of Neuroscience) which used a data-driven approach to discover a puzzling heterogeneity of patterns (including both CCN and DMN) overlapping within the dorsal PCC, but which (unlike the current manuscript) we didn't have the temporal/spatial resolution to disentangle.

*Reviewer #3 (Recommendations for the authors):*

Considering the major concern raised in the public review, I have the following suggestions to increase transparency and possibly adjust conclusions about vPCC:

1. The statistical analysis of BBG in the ventral electrodes should be reported and vPCC electrodes should be indicated in figure 3.

2. Include an analysis of the vPCC microwire probe.

3. In case the statistics and locations of vPCC electrodes do not support the above-mentioned conclusion, I recommend that it should be qualified accordingly to include a statement about the vPCC data not being completely conclusive and that further complementary research within this region is warranted.

Further, I want to express that this kind of research is crucial right now for neuroscience moving forward. I have read about similar anatomical subdivisions within the precuneus, which also gets thrown into the DMN-bucket often (at least in human fMRI), even though anatomical connections and functional connectivity with both CCN and sensory systems clearly suggest a more diverse set of functions. I also see that you have data from many more electrodes, possibly within the precuneus and I suggest you to take a look at those (in case you are not doing so already) with a similar hypothesis.

---

## [Author Response]

Essential revisions:1. Why was only broadband γ considered? I think this is fine, but in terms of relating to other datasets (especially extra-cranial) does limit the interpretation somewhat.

As a key impetus for our work was prior single-unit findings from the non-human primate, we sought to focus on spiking-related activities within the human posterior cingulate cortex (PCC). Following a large human intracranial literature (Lachaux, Axmacher et al., 2012, Miller, Honey et al., 2014), we therefore focused local field potential (LFP) analyses on broadband γ, as a well-known correlate of population spiking activity (Manning, Jacobs et al., 2009, Ray and Maunsell 2011). In addition, broadband γ is a robust correlate of BOLD fMRI responses (Mukamel, Gelbard et al., 2005, Nir, Fisch et al., 2007). Importantly, broadband γ activity, as a coarse spiking correlate, is spatially localized and invariant across cortical regions. In contrast, low-frequency oscillations are more spatially coherent and differ across brain regions, with more varied links to other neural measures of interest (spiking / BOLD). This limits our ability to reliably identify and dissociate functional responses within PCC. In future work, we hope to examine specific frequency bands, particularly in the theta range, as part of planned studies focused on the hippocampus and PCC.

2. With the clustering, I was slightly unsure how this was performed; it would be useful to have more detail in terms of how the similarity matrix was generated. Also, with the statistics, comparing tasks/clusters probably needs some cross-validation, since the clustering is performed on task features (I think) and will bias towards finding significant differences in the ANOVA.

Thank you for raising this point and analysis suggestion. The goal of the clustering analysis was to quantify any functional response similarities across isolated PCC units. Therefore, to perform this clustering, the mean instantaneous firing rate (% change) during each of the 5 task conditions (from cue to 3 seconds post-cue) was concatenated for each unit. This resulted in 91 task response feature vectors, which captured both condition selectivity and temporal response profile of each unit (example concatenated responses are shown in Figure 4 —figure supplement 1B). These response vectors/features were then subject to standard unsupervised clustering whereby: (i) all response vectors were correlated to produce a similarity matrix (Figure 4 —figure supplement 1C); (ii) a distance matrix (1 – correlation) was then created and subject to agglomerative hierarchical clustering; (iii) the resulting clustering was then visualized via a dendrogram (Figure 4 —figure supplement 1D). We have insured that these details are clearly stated in Methods – Functional Grouping of Single Unit Activity, which now reads:

“Subsequent inspection of the individual unit event-related activity revealed a diversity of functional response profiles and differing basal firing rates (effectively cancelling condition specific effects when averaged; see Figure 4 —figure supplement 1). We therefore normalized basal firing rate differences by converting instantaneous firing rate from Hz to percent change relative to the pre-cue period, as done for the LFP data. Next, we concatenated the mean instantaneous firing rate (% change) for each task condition (from cue to 3 s post-cue) to create a task response feature vector for each unit (i.e. for each unit a time series was constructed by concatenating the mean responses over the first 3 seconds of each condition). This vector captures both condition selectivity as well as response time course for each unit. To identify any putative similarity between the functional responses of units, we performed unsupervised hierarchical clustering, as follows. First, all concatenated unit task response vectors noted above were correlated with each other to create a correlation or similarity matrix. Next, correlation values were inverted (1 – r) to obtain a distance matrix that was then subject to agglomerative hierarchical clustering (via linkage function in MATLAB). The resulting cluster tree or dendrogram was then inspected and revealed four main cluster branches (based on 90% max distance cut-off; Figure 4 —figure supplement 1D). Units were then grouped in these clusters to define functional Types 1-4. It is important to note, this unsupervised approach does not require specification of cluster number.”

Regarding statistical analysis, an ANOVA was performed to examine the influence of task, cluster type, and their interaction, on PCC unit responses as a means to capture the putative functional differences of cluster types (i.e. a significant interaction of task x cluster type). The reviewer correctly notes that clustering will seek to maximize differences, which may influence the ANOVA. However, the clustering will not impact the main effect of task (which is non-significant), but specifically the main effect of cluster type and its interaction with task. The success of clustering will in turn depend on the properties of the similarity matrix. Given that hierarchical clustering is exhaustive (i.e. no seed or pre-defined cluster number), cross-validation or randomization are useful means for assessing the consistency of cluster membership across units. We therefore re-performed hierarchical clustering via a complete leave-one-out process to quantify the rate at which units are assigned to the same cluster. This analysis showed a 94% match rate (i.e. 84/90 iterations showed the exact same unit cluster assignments as the original clustering). We have updated Results – Task Selectivity of Single Units in Dorsal PCC and Methods – Functional Grouping of Single Unit Activity with this information, which now read:

Results: “To quantify the consistency of cluster membership, we performed hierarchical clustering through a complete leave-one-out process, resulting in a 94% identical cluster rate.”

Methods: “To assess the consistency of cluster membership across units, we repeated the above clustering while iterating through a complete leave-one-out process (90 iterations). This resulted in a 94% (84/90) identical cluster rate.”

3. What is the spatial distribution of the different clusters? Is this something that can be plotted? If so, it would be potentially very interesting.

We agree and were also interested to see if single unit responses showed any spatial organization. However, the smaller number of successful single-unit recording sites (6 probes) is insufficient for assessing any spatial organization (note: while each probe contains 8-microwires, these are effectively at the ‘same’ spatial location). As noted in the *Methods* and *Results*, each cluster type was observed at most recording sites (Type 1 = 4/6; Type 2 = 6/6; Type 3 = 5/6; Type 4 = 4/6). This outcome limits concerns that a given functional type may derive from a single location or subject and suggests a greater degree of heterogeneity of the cellular level, but no strong claims about specific organization at this scale can be made given the paucity of spatial sampling. It will be important to see what spatial organization is observed at the cellular level as we obtain more single unit data from this region.

4. The statistical analysis of BBG in the ventral electrodes should be reported and vPCC electrodes should be indicated in Figure 3.

In this manuscript we chose to focus our analyses specifically on dPCC (29 recording sites) since our sampling of vPCC was far less extensive (vPCC = 6 recording sites) – as originally stated (Lines 123-4). Since mnemonic and default-network type responses in vPCC have been previously shown in neuroimaging and our own human electrophysiological data (Reviewed in Fox, Foster et al., 2018), we instead use our vPCC findings as an observational account that our recordings from this region are consistent with this prior work. Nevertheless, we have added the requested statistical analysis of BBG in vPCC electrodes to Results – Task Selectivity of LFP in dPCC. Notably, a mixed effects model revealed a significant main effect of task condition on BBG amplitude during the late task execution, with BBG amplitude significantly higher during the past condition than all other conditions, consistent with prior studies. Also, as suggested, ventral electrodes are now indicated in Figure 3 with a dashed outline (note: dorsal/ventral determinations were made based on individual anatomy, not group normalized space). Updated Results text now reads as follows:

“In contrast, sites within vPCC (n_probes_ = 6) showed the converse BBG response profile across conditions, consistent with the DMN and prior observations (Dastjerdi, Foster et al., 2011, Foster, Dastjerdi et al., 2012, Foster, Rangarajan et al., 2015), where BBG responses were greater for the episodic task conditions (see Figure 2 —figure supplement 1). A mixed effects model revealed a significant main effect of condition on BBG amplitude, with vPCC mean BBG amplitude being significantly higher for past than all other conditions (past-add: t(1253) = 8.841, p < 0.0001; past-search: t(1253) = 5.151, p < 0.0001; past-rest: t(1253) = 3.978, p = 0.0007; past-future: t(1253) = 3.020, p = 0.0216), and significantly lower for add than all other conditions (add-past: t(1253) = -8.841, p < 0.0001; add-future: t(1253) = -5.811, p < 0.0001; add-rest: t(1253) = -4.863, p < 0.0001; add-search: t(1253) = -3.690, p = 0.0022).”

5. Please include an analysis of the vPCC microwire probe.

Following our criteria detailed in the *Methods*, we pursued analysis of the vPCC microwire probe and determined that it failed to reliably detect any isolated single unit activity.

6. In case the statistics and locations of vPCC electrodes do not support the above-mentioned conclusion, I recommend that it should be qualified accordingly to include a statement about the vPCC data not being completely conclusive and that further complementary research within this region is warranted.

In line with our response to comment 4, this manuscript is primarily focused on establishing the selectivity of dPCC populations and units, coupled with an observational account of vPCC mnemonic/default activity that is consistent with prior work in human neuroimaging and electrophysiology (Fox, Foster et al., 2018). The addition of vPCC statistics consistent with prior literature, in combination with our robust dPCC results, further support our conclusions regarding putative functional organization of PCC. We agree with the reviewer that the limitations of a small vPCC sample size, and its precedence in the literature, should be more clearly addressed in the Discussion. Therefore, we have added remarks that qualify the scope of our data, noting that more extensive sampling will be necessary to fully determine the complex functional neuroanatomy of PCC, but that our findings replicate and extent prior work. The text reads as follows:

“Regarding the present study, we note that our focus was on dPCC, with vPCC being more sparsely sampled, limiting a full characterization of functional gradients. However, we also note that our vPCC findings are consistent with a number of intracranial studies showing DMN-like responses within vPCC (Fox, Foster et al., 2018).”

7. Participants were excluded if they presented with frank anatomical abnormalities, however, it is not clear how this decision was made. Even if gross anatomical abnormalities are not evident, subtle changes in structural and functional connectivity might be present in this group. I wonder if the authors might therefore comment on how representative these results are to inform the functional architecture of the PCC and its subdivisions in general.

We agree that it is important to consider and attempt to reduce the potential impact of pathology-related differences between subjects in our study and the general population. Therefore, multiple steps are taken to ensure that the data used represents typical electrical activity and cognitive functioning. All subjects perform neuropsychological evaluations prior to testing to determine if cognitive functioning is typical and no overt deficiencies are present. We do not move forward with collecting task data in individuals with such deficits (e.g. poor memory). Further, we do not consider subjects with prior lesions (clinical or surgical) or pathological activity observed in PCC. Indeed, we exclude subjects with prior lesions to any brain structure. This is based on extensive structural and tractographic imaging generated as part of the patient’s clinical work up. Ultimately, while subtle changes cannot be fully precluded, the generalizability of human intracranial results is supported by its consistency with fMRI data in neurotypical subjects, as well as electrophysiological results in non-human primates. Indeed, we hope our findings serve to highlight and integrate the consistencies between prior neuroimaging and electrophysiological results across species.

References

Dastjerdi, M., B. L. Foster, S. Nasrullah, A. M. Rauschecker, R. F. Dougherty, J. D. Townsend, C. Chang, M. D. Greicius, V. Menon, D. P. Kennedy and J. Parvizi (2011). "Differential electrophysiological response during rest, self-referential, and non-self-referential tasks in human posteromedial cortex." Proc Natl Acad Sci U S A 108(7): 3023-3028.

Foster, B. L., M. Dastjerdi and J. Parvizi (2012). "Neural populations in human posteromedial cortex display opposing responses during memory and numerical processing." Proc Natl Acad Sci U S A 109(38): 15514-15519.

Foster, B. L., V. Rangarajan, W. R. Shirer and J. Parvizi (2015). "Intrinsic and task-dependent coupling of neuronal population activity in human parietal cortex." Neuron 86(2): 578-590.

Fox, K. C. R., B. L. Foster, A. Kucyi, A. L. Daitch and J. Parvizi (2018). "Intracranial Electrophysiology of the Human Default Network." Trends Cogn Sci 22(4): 307-324.

Lachaux, J. P., N. Axmacher, F. Mormann, E. Halgren and N. E. Crone (2012). "High-frequency neural activity and human cognition: past, present and possible future of intracranial EEG research." Prog Neurobiol 98(3): 279-301.

Leech, R., R. Braga and D. J. Sharp (2012). "Echoes of the brain within the posterior cingulate cortex." J Neurosci 32(1): 215-222.

Leech, R., S. Kamourieh, C. F. Beckmann and D. J. Sharp (2011). "Fractionating the default mode network: distinct contributions of the ventral and dorsal posterior cingulate cortex to cognitive control." J Neurosci 31(9): 3217-3224.

Manning, J. R., J. Jacobs, I. Fried and M. J. Kahana (2009). "Broadband shifts in local field potential power spectra are correlated with single-neuron spiking in humans." J Neurosci 29(43): 13613-13620.

Miller, K. J., C. J. Honey, D. Hermes, R. P. Rao, M. denNijs and J. G. Ojemann (2014). "Broadband changes in the cortical surface potential track activation of functionally diverse neuronal populations." Neuroimage 85 Pt 2: 711-720.

Mukamel, R., H. Gelbard, A. Arieli, U. Hasson, I. Fried and R. Malach (2005). "Coupling between neuronal firing, field potentials, and FMRI in human auditory cortex." Science 309(5736): 951-954.

Nir, Y., L. Fisch, R. Mukamel, H. Gelbard-Sagiv, A. Arieli, I. Fried and R. Malach (2007). "Coupling between neuronal firing rate, γ LFP, and BOLD fMRI is related to interneuronal correlations." Curr Biol 17(15): 1275-1285.

Ray, S. and J. H. Maunsell (2011). "Different origins of γ rhythm and high-γ activity in macaque visual cortex." PLoS Biol 9(4): e1000610.